# Improving the Separation Properties of Polybenzimidazole Membranes by Adding Acetonitrile for Organic Solvent Nanofiltration

**DOI:** 10.3390/membranes13010104

**Published:** 2023-01-12

**Authors:** Ga Yeon Won, Ahrumi Park, Youngmin Yoo, You-In Park, Jung-Hyun Lee, In-Chul Kim, Young Hoon Cho, Hosik Park

**Affiliations:** 1Green Carbon Research Center, Chemical Process Division, Korea Research Institute of Chemical Technology, Daejeon 34114, Republic of Korea; 2Department of Chemical and Biological Engineering, Korea University, Seoul 02841, Republic of Korea; 3Department of Advanced Materials and Chemical Engineering, University of Science and Technology, Daejeon 34113, Republic of Korea

**Keywords:** organic solvent nanofiltration, polybenzimidazole, nonsolvent-induced phase separation, co-solvent, evaporation, crosslink

## Abstract

In research on membranes, the addition of co-solvents to the polymer dope solution is a common method for tuning the morphology and separation performance. For organic solvent nanofiltration (OSN) applications, we synthesized polybenzimidazole (PBI) membranes with high separation properties and stability by adding acetonitrile (MeCN) to the dope solution, followed by crosslinking with dibromo-p-xylene. Accordingly, changes in the membrane structure and separation properties were investigated when MeCN was added. PBI/MeCN membranes with a dense and thick active layer and narrow finger-like macrovoids exhibited superior rejection properties in the ethanol solution compared with the pristine PBI membrane. After crosslinking, they displayed superior rejection properties (96.56% rejection of 366-g/mol polypropylene glycol). In addition, the membranes demonstrated stable permeances for various organic solvents, including acetone, methanol, ethanol, toluene, and isopropyl alcohol. Furthermore, to evaluate the feasibility of the modified PBI OSN membranes, ecamsule, a chemical product in the fine chemical industry, was recovered. Correspondingly, the efficient recovery of ecamsule from a toluene/methanol solution using the OSN process with PBI/MeCN membranes demonstrated their applicability in many fine chemical industries.

## 1. Introduction

Organic solvent nanofiltration (OSN) is a pressure-driven membrane filtration technology that separates solutes dissolved in various organic solvents. The separation mechanism of OSN is based on size exclusion, and the membrane has a molecular weight cut-off (MWCO) between 200 and 1000 g/mol [1,2]. Conventional separation technologies, including distillation, evaporation, chromatography, and extraction, are energy- and cost-intensive, generate hazardous wastes, and degrade heat-sensitive products during distillation and evaporation at high temperatures [3]. Accordingly, OSN has been studied as a suitable alternative to traditional separation technologies, such as solute separation [4], solvent recovery [5,6], purification [7], extraction and concentration, because of the numerous advantages of OSN. These include: (1) simple pressure-driven process; (2) cost and energy efficiency; and (3) reduced thermal damage to high-value chemical products [8]. OSN application studies, such as refining active pharmaceutical ingredients and genotoxic impurities and removing free fatty acids generated in food production, have demonstrated the feasibility of OSN processes in various industries, including the pharmaceutical, food, petrochemical, and fine chemical sectors [4,6,9,10].

An OSN membrane requires an excellent separation performance and high chemical stability under harsh environmental conditions in various chemical industries. The outstanding separation features of the OSN membrane, such as high rejection, permeance, and long-term stability, and the sturdiness of the materials in various solvents, including polar, acidic, and basic solvents, are of primary interest. For OSN membranes, numerous polymeric materials, such as polyimide [11,12], poly(ether ketone) [13,14], polyamide [15,16], and polybenzimidazole (PBI) [17,18], have been explored. These polymers, which contain aromatic or imide groups in their chemical structure, are stable in various organic solvents [19]. Among them, PBI, an aromatic polymer with strong hydrogen bonding with benzimidazole groups [8,20], has been used as a membrane material in several membrane separation processes, including gas separation [21], forward osmosis [22], and nanofiltration for water treatment [23]. In addition, PBI has been explored for use in OSN owing to its exceptional chemical, thermal, and physical stability [24].

For OSN applications, integrally skinned asymmetric PBI-based membranes are fabricated via the nonsolvent-induced phase separation (NIPS) method. The membrane structure influences the separation qualities, such as solute rejection and solvent permeability. Adding co-solvents or additives to polymer dope solutions is a typical technique for controlling the membrane structure during NIPS fabrication. Chen et al. fabricated a PBI membrane by adding a volatile solvent and showed an increase in the active layer thickness and selectivity of the PBI membrane [25]. When a volatile solvent is added as a co-solvent, the solvent on the surface of the membrane evaporates to generate a polymer-rich phase before immersion in the coagulation bath. This results in the formation of a dense and thick active layer, which enhances solute rejection [26]. Wang et al. fabricated a poly(m-phenylene isophthalamide) hollow fiber membrane by adding acetone as a volatile non-solvent into the spinning solution; the membrane exhibited a structure with reduced macrovoids and enhanced salt rejection [27]. When a co-solvent with a good affinity with the polymer is added to the polymer dope solution, the viscosity of the dope solution decreases [28], which makes it difficult to prepare the mechanically robust membrane. However, as the solvent with low affinity with polymer is added to the polymer dope solution, the polymer solubility of the solvent decreases, and the interaction between the polymer chains increases. As a result, the polymer chain assumes a tightly coiled configuration, preventing the solution in the polymer chain from diffusing to the coagulation solvent and exhibiting a membrane structure with macrovoid suppression [29,30].

Although the membrane is fabricated using a polymer with high chemical and physical stability, most polymer membranes prepared using NIPS may be dissolved in polar aprotic organic solvents, including *N*,*N*-dimethylacetamide (DMAc), *N*,*N*-dimetyformamide, and *N*-methyl pyrrolidone. Owing to the need for a process that increases the chemical and physical stability of the membrane for use in the chemical industry with various organic solvents, various crosslinking methods for the NIPS-based polymeric membrane have been documented. PBI membranes with an amine group can be crosslinked using a variety of crosslinkers, including glutaraldehyde, 1,2,7,8-diepoxyoctane, sulfuric acid, and trimesoyl chloride [20,31,32,33]. Some crosslinked PBI membranes have exhibited an increased separation performance and chemical resistance, but they are difficult to apply to the OSN field due to their solubility in organic solvents and limited permeability [34]. Among the various crosslinkers, PBI membranes crosslinked with dibromo-p-xylene (DBX) demonstrated high chemical stability in polar aprotic solvents and an enhanced separation performance, making them suitable for OSN applications [35,36].

In this study, we demonstrate a PBI membrane with both a better separation performance and improved mechanical stability, fabricated through acetonitrile addition and crosslinking. The effect of a solvent additive with a low affinity with PBI on the membrane structure and separation properties has not been reported. Accordingly, we synthesized PBI membranes via NIPS using acetonitrile (MeCN), a volatile solvent with a low affinity for PBI, as a co-solvent. The membranes were then subjected to nanofiltration crosslinked with DBX to improve their chemical stability and separation properties. To make the membranes practically usable, they were subjected to a nanofiltration test using an ecamsule solution, from which ecamsule was rejected.

## 2. Materials and Methods

### 2.1. Materials

To fabricate the membranes, Celazole S26 (PBI Performance Products, Charlotte, NC, USA) was used. It contained 25.8 wt% PBI and 1.5 wt% lithium chloride in DMAc. DMAc (99.5% Samchun Co., Pyeongtaek, Republic of Korea) and acetonitrile (MeCN, 99.5%, Duksan C&P, Daejeon, Republic of Korea) were used as solvent to dissolve PBI. Polypropylene (PP) nonwoven (Novatexx 2471, Freudenberg Filtration Technologies, Weinheim, Germany) was used as a support layer. The fabricated membranes were stored in isopropyl alcohol (IPA, 99.5%, Duksan C&P, Daejeon, Republic of Korea). As a crosslinker, $\alpha ,\alpha^{\prime }$-dibromo-p-xylene (DBX, 97%, Sigma-Aldrich Co., St. Louis, MO, USA) was used. For the OSN performance test, anhydrous ethyl alcohol, acetone, methanol, and toluene were purchased from Duksan C&P (Daejeon, Republic of Korea); poly(propylene) glycol (PPG) (average Mn: 425, 725, and 1000 g/mol) was purchased from Sigma-Aldrich (St. Louis, MO, USA). The ecamsule solution was provided by Withel (Iksan, Republic of Korea).

### 2.2. Membrane Fabrication

The PBI membranes were fabricated using the NIPS technique. First, the Celazole S26 PBI solution was diluted with DMAc and MeCN. Then, the PBI dope solution was stirred for 6 h and used after 12 h of stabilization. The fabrication of the PBI membrane was conducted in an enclosed room with a temperature of 22 °C and humidity of 30–35%. The PBI dope solution was cast on a PP support using an automatic film applicator at 75 mm/s with a casting thickness of 200 µm. The PBI film had an evaporation time of 20 s; it was submerged in a deionized water bath to coagulate for 1 h and left for 24 h in an exchanged deionized water. The PBI membranes were then rinsed with, and stored in, IPA.

To crosslink the fabricated PBI membranes, a DBX/MeCN solution was prepared by dissolving 3 wt% of DBX in MeCN for 1 h at 80 °C. Accordingly, the PBI membranes were immersed and stirred in a prepared DBX/MeCN solution at 80 °C for crosslinking. After 24 h, the crosslinked PBI membranes were washed for 1 h with IPA and then stored in IPA until use. The dope compositions of the membranes developed for this study are summarized in Table 1.

### 2.3. Membrane Characterization

The viscosity of the dope solutions containing varying concentrations of MeCN was determined using a Brookfield DV_2_ T^®^ Viscometer and SC4-21 Spindle (Brookfield Ametek, Middleboro, MA, USA). The tests were performed at 22 °C, the temperature at which the membrane was fabricated, at a spindle speed of 10 rpm. To observe the membrane morphology, environmental scanning electron microscopy (E-SEM, Quattro S, Thermo Fisher Co., Waltham, MA, USA) was used. The membranes were dried in a vacuum oven and sliced in liquid nitrogen before being coated with platinum using a sputtering device (JEOL JFC-1300, Tokyo, Japan). The chemical structure of the membranes before and after crosslinking were investigated and confirmed via Fourier transform infrared spectroscopy (FT-IR, Nicolet 5700, Thermo Fisher Co., Waltham, MA, USA) in the wavelength range of 400–4000 cm^−1^ with 32 scans for each membrane and X-ray photoelectron spectroscopy (XPS, K-Alpha+, Thermo Fisher Co., Waltham, MA, USA).

### 2.4. OSN Test

To evaluate the OSN performance of the membranes, both the dead-end and crossflow filtration systems, illustrated in Figure 1, were used. Before the test, the membranes were submerged for 1 h in a filtering solvent to remove any leftover IPA. The effective area of the membrane used in the crossflow system was 14.8 cm^2^. Three distinct PPG products with various average molecular weights (~400, ~725, and ~1000 g/mol) were dissolved in ethanol at a concentration of 1 g/L. The filtration was performed at 30 °C and 15 bar. For the dead-end filtration systems, the effective area was 19.6 cm^2^, stirred filtration was performed at 15 bar, and the stirrer speed is determined as 250 rpm.

In both filtration systems, the solution permeance (J, L m^−2^ h^−1^ bar^−1^) and solute rejection (R, %) were calculated by Equations (1) and (2), respectively:(1)J=V/AtΔP
(2)$$R=\left(1-c_{p}/c_{f}\right)\times 100$$
where V(L) is the permeated volume through the membrane, A(m^2^) is the effective area of the membrane, t(h) is permeated time, and ΔP the operating pressure (bar). C_p_ and C_f_ are the concentrations of the permeated solution and feed solution (g/L), respectively. The C_p_ and C_f_ were analyzed via high-performance liquid chromatography (1260 Infinity II, Agilent Technologies Co., Santa Clara, CA, USA) and an evaporative light scattering detector (SEDEX 75, SEDERE Co. Olivet, France). The columns used in the high-performance liquid chromatography are the Kinetex^®^ C_18_ Column (50 × 4.6 mm, 2.6-μm particle size, Phenomenex Co., Torrance, CA, USA) for PPG analysis and the Poroshell 120 EC-C_18_ Column (150 × 4.6 mm, 4-μm particle size, Agilent Technologies Co., Santa Clara, CA, USA) for ecamsule solution analysis.

## 3. Results and Discussion

### 3.1. Effects of MeCN Concentration

#### 3.1.1. Membrane Morphology

Table 2 shows the solubility parameters and boiling point of the PBI and the solvents. The solubility parameters consider three main types of interaction between molecules: dispersion forces (δ_d_), polar forces (δ_p_), and hydrogen bonding forces (δ_h_). The affinity of the solvents with polymer was evaluated using R_HSP_, which represents a distance in the Hansen solubility parameters between solvent (A) and polymer (B). R_HSP_ is calculated by the following Equation (3):(3)$$R_{HSP}=\sqrt{4{\left(\partial_{dA}-\partial_{dB}\right)}^{2}+{\left(\partial_{pA}-\partial_{pB}\right)}^{2}+{\left(\partial_{hA}-\partial_{hB}\right)}^{2}}$$

Increased R_HSP_ values indicate a reduced interaction between the solvent and polymer [40]. Compared with DMAc, the $R_{HSP}$ between MeCN and PBI was higher, indicating a low affinity of MeCN with PBI. In a mixed solvent of DMAc and MeCN, the PBI solubility decreased, thus increasing the interactions between the PBI polymer chains. These increasing interactions resulted in a tightly coiled conformation of the PBI polymer chains, thereby increasing the viscosity of the PBI dope solution (Figure 2) [30,41]. By decreasing the PBI solubility in the mixed solvent with further MeCN addition, the PBI hardly dissolved in a solvent including 25 wt% MeCN. Therefore, the PBI dope solution was prepared with a solvent containing up to 20 wt% of MeCN. The PBI dope solution with low viscosity permeated further into the PP support as the evaporation time increased during the evaporation process after casting; therefore, the PBI membrane formed by giving an evaporation time of 20 s. The morphology of the PBI membrane was observed through E-SEM.

Figure 3 shows the cross-sectional SEM images of the membranes. The thickness of the PBI membrane and the active layer increased with the increasing MeCN concentration. The cross-sectional image of M1 revealed numerous macrovoids and a very thin active layer (0.31 μm). In contrast, M4 has a dense structure, narrow macrovoids, and a thick active layer (1.95 μm). Before immersing the membrane in the coagulation bath, the solvent on the membrane surface is evaporated, thereby forming a polymer-rich phase [42]. As shown in Table 2, MeCN is a more volatile solvent than DMAc. As the concentration of MeCN increased, the solvent on the surface of the membrane evaporated more, resulting in a thicker and more compact active layer.

Due to the high viscosity of PBI, the attraction between the PBI molecules is strong. As shown in Figure 2 and Figure 3, the increased MeCN concentration led to an increase in the viscosity of the PBI dope solution, thus increasing the total thickness of the PBI. Due to the strong attraction between the PBI chains, in the process of casting the PBI solution onto the PP support, the penetration of the PBI solution into the PP support was slowed and the total thickness increased. In addition, compared with M1, the PBI/MeCN membranes possessed narrower macrovoids and exhibited sponge-like porosity morphologies toward the skin layer. Owing to the increased viscosity of the PBI solution and the tighter and thicker surface layer caused by the evaporation of the volatile solvent, the solvent in the membrane was prevented from diffusing into the nonsolvent in the coagulation bath, and the demixing between the solvent and nonsolvent was delayed, resulting in the formation of a membrane with a sponge-like structure [43].

#### 3.1.2. OSN Performance of PBI Membranes

Figure 4 presents the results of the ethanol and ethanol–PPG solution filtration experiments using a crossflow filtration system. M1 displayed the highest permeance and the lowest rejection qualities, whereas M4 demonstrated the lowest permeance and highest rejection qualities. As the MeCN concentration increased, the ethanol permeance of the membrane decreased from 10.22 $Lm^{-2}h^{-1}{bar}^{-1}$ to 3.76 $Lm^{-2}h^{-1}{bar}^{-1}$. From M1 to M4, the rejection of PPG 366 g/mol increased from 23.03% to 65.93%. The membrane formed by the immediate demixing of the solvents and non-solvents had a thin, porous surface layer, whereas the membrane formed by the delayed demixing of the solvents and non-solvents had a dense, thick surface layer [44]. The addition of MeCN delayed the demixing of the solvents and non-solvents and, as shown in Figure 3, the membranes fabricated with dope solutions with a higher MeCN concentration had thick and compact active layers. As the thick and dense outermost layer of the membrane increased the solvent’s permeation resistance, the ethanol permeance decreased as the MeCN concentration increased. Further, because the evaporation of MeCN resulted in a denser membrane surface, the rejection property of the membrane was significantly strengthened. The MWCO of the M1 was >1000 g/mol. The increased MeCN concentration increased the PPG rejection; M3 had an MWCO of 949.4 g/mol and M4 had an MWCO of 716.2 g/mol.

### 3.2. Crosslinking Effect on Membrane Properties

#### 3.2.1. FT-IR/XPS Analysis

Crosslinking the membranes with DBX improved their separation properties and chemical stability. The FT-IR investigation confirmed that DBX reacted with the amine group in the PBI imidazole rings (Figure 5a) [34,36]; the FT-IR spectra of the membrane before and after crosslinking are shown in Figure 5b. In addition, –NH–, –N=C–, and N–H were attributed to the remarkable alterations following crosslinking, detected in the peak at 1285 cm^−1^ and the broadband at 1440 cm^−1^, which appeared at ~3150 cm^−1^ [45]. Compared with the spectra of M1, the intensity of the –N=C– transmittance peak decreased, whereas the –NH– peak and the N–H band disappeared in the M1-X spectra. This phenomenon was a result of the formation of a new covalent bond between the nitrogen functional groups in PBI and the Bromo functional groups in DBX.

In addition, the XPS analysis confirmed the chemical composition of the surface of the crosslinked PBI membrane. As shown in Figure 5c, the M1 spectra contain both pyrrolic and pyridinic nitrogen atom signals, which are attributed to the amine group and the imine of PBI. Following crosslinking, the concentration of pyrrolic and pyridinic nitrogen decreased. The area ratio of the pyrrolic peak decreased from 60.29% to 13.04%, whereas that of the pyridinic peak decreased from 39.71% to 21.75%. The reaction of the pyrrolic and pyridinic nitrogen atoms in the amine and imine group of PBI with bromine in DBX resulted in the formation of a new peak containing a large number of graphitic nitrogen atoms in the M1-X spectra. The presence of graphitic nitrogen atoms suggests that the PBI membrane was successfully crosslinked with DBX.

#### 3.2.2. Membrane Morphology

The cross-sectional SEM images of the crosslinked membrane are displayed in Figure 6. Compared with the pristine membranes (Figure 3), the crosslinked membranes exhibited a narrowing of the macrovoids and an increase in the total membrane thickness. In addition, as illustrated in Figure 6e–h, the active layer on the surface of the membrane became thicker. Following crosslinking with DBX, the active layer increased from a minimum of 1.57 times to a maximum of 2 times, and the total thickness increased from a minimum of 1.35 to a maximum of 1.98 times. These increases are a result of the MeCN membrane swelling. The membranes were immersed in a DBX/MeCN solution, resulting in membrane expansion. The PBI polymer chains were crosslinked with DBX to form a network structure, increase membrane thickness, and preserve the expanded membrane. In the case of the dense active layer, in which the polymer concentration was more than that of the sublayer, a greater volume expansion was observed along with a marginally higher growth rate [46].

#### 3.2.3. OSN Performance of Crosslinked Membranes

Figure 7 shows the separation performance, including the pure solvent permeance and PPG (dissolved in ethanol) rejections, measured at 15 bar. Compared with the pristine membranes, all of the crosslinked membranes demonstrated decreased ethanol permeance and increased PPG rejection. Although crosslinking increased the separation properties of M1, the MWCO of M1-X was >1000 g/mol, making its nanofiltration use challenging. M3 demonstrated a pure ethanol solution permeance of 4.49 $Lm^{-2}h^{-1}{bar}^{-1}$. After crosslinking, the pure ethanol permeance of M3-X was 3.41 $Lm^{-2}h^{-1}{bar}^{-1}$. The MWCO of the crosslinked M3 membrane decreased from 949.4 g/mol to 774.5 g/mol in the ethanol–PPG solution. The M4-X with the thickest active layer exhibited the lowest ethanol permeance and the lowest MWCO, which was <366 g/mol. Through crosslinking, the generated polymer network decreased the flexibility of the PBI polymer chain and reduced the pore size, thus making it more difficult for the solvent to permeate [47] and decreasing the ethanol permeance of the membranes. Simultaneously, the rejection of solutes increased due to the reduction in the pore size by the crosslinked three-dimensional polymer network structure [34]. Thus, the crosslinked membranes had a lower MWCO range than the pristine membrane, enhancing their selectivity. In Table 3, the OSN performance between the PBI-based membranes fabricated in this study and the commercial OSN membranes are listed for a comparison of the membranes’ performance. The M4-X shows an excellent separation performance and a much higher rejection for low molecular weight solutes than most of the reported OSN membranes due to the tight and thick PBI/MeCN selective layer.

Long-term nanofiltration experiments with an ethanol-PPG solution using the M1-X and M4-X were conducted for 96 h at 15 bar and are illustrated in Figure 7c,d. For 96 h, the solution permeance of the M1-X declined by approximately 43%. However, the M4-X showed over 95% of the rejection for 366 g/mol PPG, and the solution permeance decreased from 2.14 $Lm^{-2}h^{-1}{bar}^{-1}$ to 1.73 $Lm^{-2}h^{-1}{bar}^{-1}$. The membranes were compressed for a long time, resulting in a decrease in the solvent permeance. As the membrane has more macrovoid in its membrane structure, the membrane has lower resistance to pressure [53,54]. The M1-X showed a loose structure even after crosslinking and showed that the permeability of the M1-X decreased significantly due to low compression resistance during long-term operation. In contrast, the M4-X had a rigid structure with narrow pore size distribution after crosslinking, showing a slight decrease in solvent permeability.

The OSN performances of the crosslinked membranes evaluated with ethanol, IPA, methanol, acetone, and toluene are shown in Figure 7e. The relationship between the permeances of pure solvents and solvent viscosity is depicted in Figure 7f. Several variables affect solvent permeation, including viscosity, solubility parameters, and molecular diameter. Among these, viscosity had a strong association with solvent permeability [48,55,56,57]. The sequence of the permeance of different solvents is as follows: acetone > methanol > ethanol > toluene > IPA. With the exception of the nonpolar toluene, the polar solvent permeances have a proportional correlation (R^2^ = 0.9758) with the reciprocal of solvent viscosity. In general, a solvent with low viscosity causes less disruption while permeating through the membrane. The permeance of acetone with the lowest viscosity was the highest, whereas that of IPA with the highest viscosity was the lowest. These findings indicate that viscosity is an important factor in solution permeability and the prediction of the permeance of PBI membranes in other polar solvents is possible.

Through crosslinking, the chemical stability of the membrane in several organic solvents was increased, as was its separation capability. To confirm that the membrane has potential applications in the chemical industry, a nanofiltration test was performed using the ecamsule solution, a strong base solution comprised of ecamsule, toluene, methanol, and tert-butoxide. Considering that the molecular weight of ecamsule is 562.69 g/mol, Figure 8 shows the results of the dead-end filtration experiment conducted at 15 bar using the M3-X and M4-X. M3-X and M4-X exhibited low permeance values of 0.83 $Lm^{-2}h^{-1}{bar}^{-1}$ and 0.52 $Lm^{-2}h^{-1}{bar}^{-1}$, respectively. M3-X rejected 77.04% of the ecamsule, whereas M4-X rejected 90.22%. Toluene and methanol present in the ecamsule solution inflated the membranes, resulting in lower rejection than the PPG rejection with a similar molecular weight, and ecamsule was deposited inside the membrane throughout the filtering process, resulting in low solvent permeability. Despite the deterioration of the separation performance, an exclusion rate of >90% was observed for strongly basic solvents without membrane defects. This indicates that the crosslinked PBI-MeCN membrane has great potential applications for actual OSN implementations of solvent purification and valuable chemical product recovery.

## 4. Conclusions

In this study, to fabricate membranes with superior separation properties, membranes were synthesized using a PBI dope solution with MeCN as a co-solvent, and they were crosslinked with DBX to improve their chemical stability and separation property for OSN application. With the addition of MeCN, solvent evaporation at the membrane surface and viscosity increased, resulting in the formation of membranes with sublayer-suppressed microvoids and a thick active layer. The OSN experiment of the membranes was conducted using ethanol and PPG in a crossflow filtering system. As the concentration of MeCN increased, the altered structure exhibited reduced permeance and enhanced PPG rejection. After crosslinking with DBX, the chemical structure of the membranes was validated via FT-IR and XPS analysis, and they were successfully crosslinked. The total and active layer thickness of the crosslinked membranes increased; however, the macrovoids shrunk due to the irreversible swelling that occurred in the crosslinking solution. All of the crosslinked membranes exhibited a superior separation performance compared with the pristine membranes. The membranes containing 20 wt% MeCN had a 2.69 $Lm^{-2}h^{-1}{bar}^{-1}$ ethanol permeance and 96.56% PPG rejection at 366 g/mol, which is a 30% improvement over the pristine membranes. Using various organic solvents, the membranes were subjected to a nanofiltration test, and the permeability of the solvents increased in the following order: acetone > methanol > ethanol > toluene > IPA. A correlation was observed between the permeability and viscosity of the solvents, except in the case of toluence, which is a nonpolar solvent.

In addition, the membranes were tested using the ecamsule solution, which is used in the fine chemical industry. The membranes had a permeance of 0.52 $Lm^{-2}h^{-1}{bar}^{-1}$ and an ecamsule rejection rate of 90.22%. The ecamsule solution nanofiltration was conducted without membrane defects, indicating that the crosslinked PBI/MeCN membranes with superior separation properties have the potential for application in solvent purification, reuse, and the recovery of the valuable chemical product, as well as indicating that additional research is required to improve the OSN performance.

## Figures and Tables

**Figure 1 membranes-13-00104-f001:**
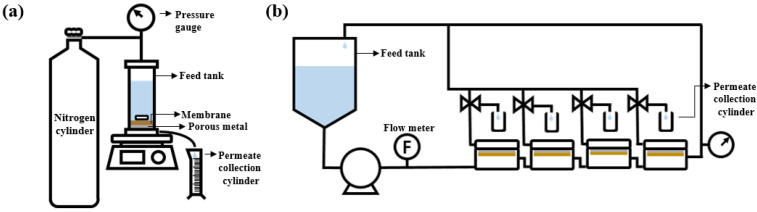
Schematic of (**a**) dead-end filtration system and (**b**) crossflow filtration system.

**Figure 2 membranes-13-00104-f002:**
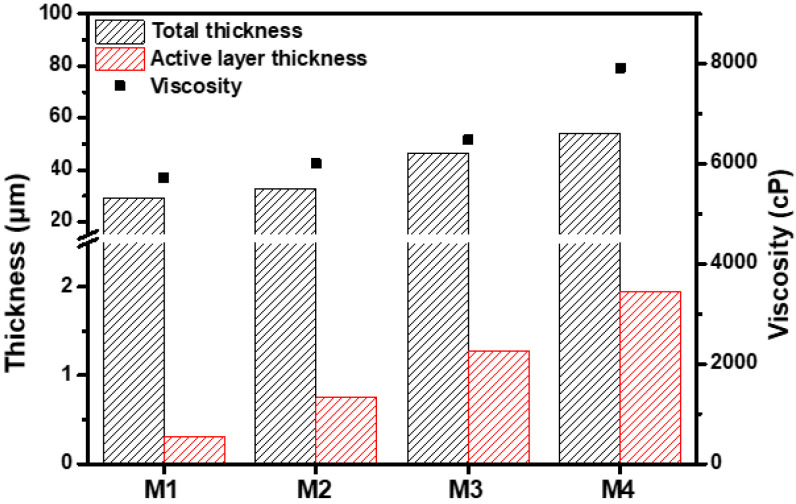
Viscosity and membrane thickness and active layer thickness of M1–M4.

**Figure 3 membranes-13-00104-f003:**
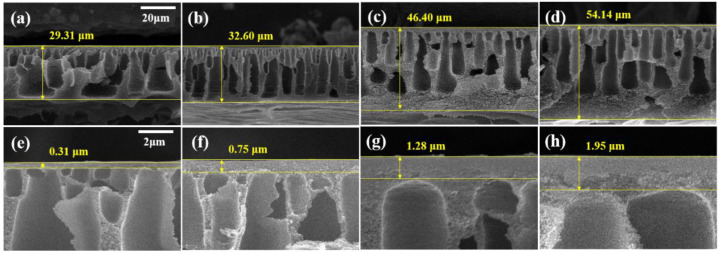
SEM images of membranes (**a**,**e**) M1, (**b**,**f**) M2, (**c**,**g**) M3, and (**d**,**h**) M4. (**a**–**d**) Cross-section SEM images with 20-μm scale bars and (**e**–**h**) enlarged cross-section images with 2-μm scale bars.

**Figure 4 membranes-13-00104-f004:**
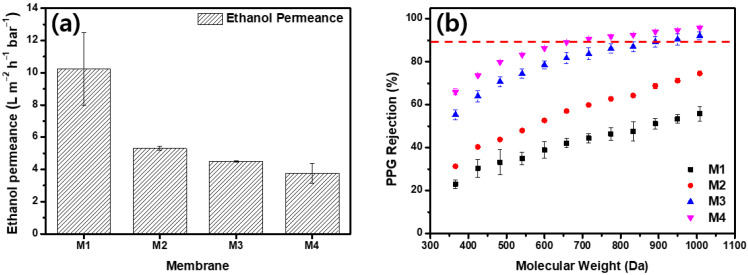
Separation performance of membranes. (**a**) Pure ethanol permeance and (**b**) polypropylene glycol (dissolved in ethanol) rejection of membranes.

**Figure 5 membranes-13-00104-f005:**
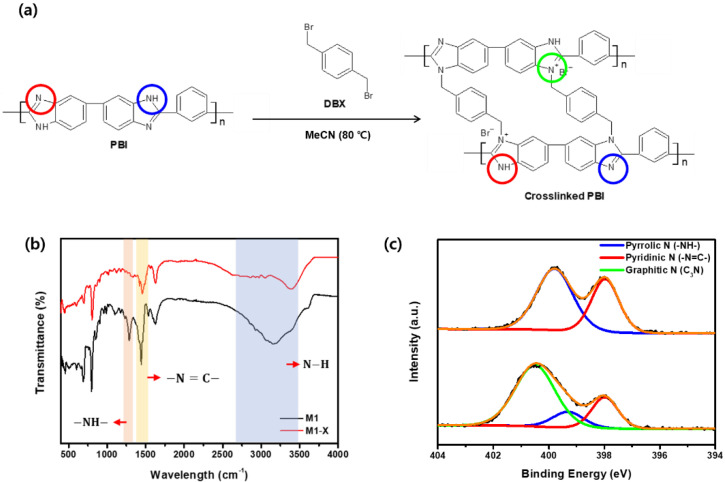
(**a**) Crosslinking reaction of PBI with DBX (red, blue, green circles mean pyridinic nitrogen, pyrrolic nitrogen, and Graphitic nitrogen, respectively) (**b**) Fourier transform infrared spectroscopy spectra and (**c**) X-ray photoelectron spectroscopy N 1 s spectra of M1 and M1-X.

**Figure 6 membranes-13-00104-f006:**
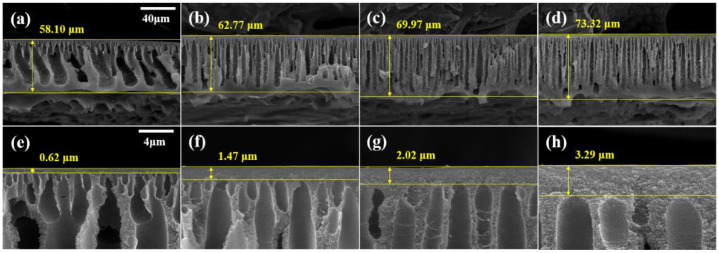
SEM images of membranes (**a**,**e**) M1-X (**b**,**f**) M2-X, (**c**,**g**) M3-X, and (**d**,**h**) M4-X. (**a**–**d**) Cross-section SEM images with 40-μm scale bars and (**e**–**h**) enlarged cross-section images with 4-μm scale bars.

**Figure 7 membranes-13-00104-f007:**
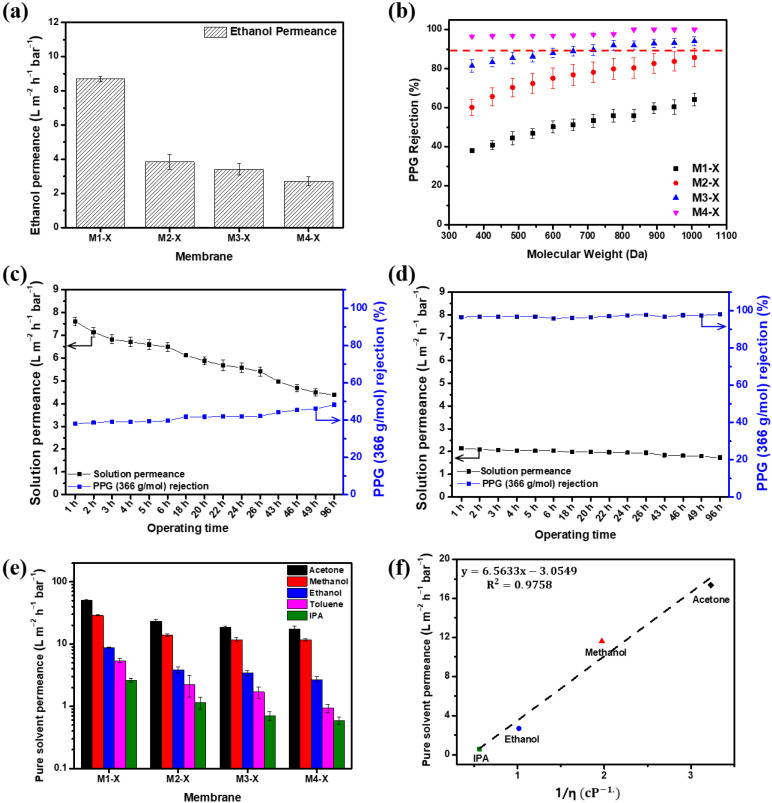
Separation performance of the membranes. (**a**) Pure ethanol permeance, (**b**) polypropylene glycol (dissolved in ethanol) rejection of membranes with crosslinking, (**c**) long-term OSN test in ethanol PPG solution of M1-X, (**d**) long-term OSN test in ethanol PPG solution of M4-X, (**e**) organic solvent permeance, and (**f**) solvent permeances test at 15 bar vs. reciprocal of viscosity (η) of solvent for M4-X.

**Figure 8 membranes-13-00104-f008:**
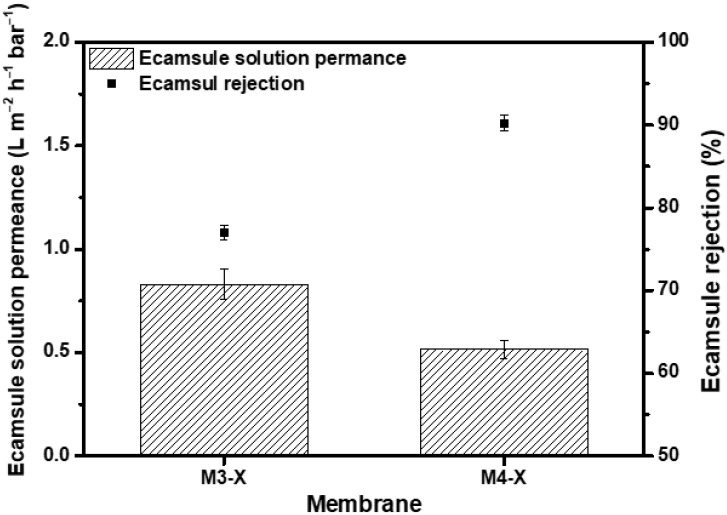
Permeance of ecamsule solution and ecamsule rejection at 15 bar.

**Table 1 membranes-13-00104-t001:** Membrane dope composition.

Membrane	Materials			DBX Crosslink
	PBI (wt%)	DMAc (wt%)	MeCN (wt%)	
M1	18	82	0	No
M1-X	18	82	0	Yes
M2	18	72	10	No
M2-X	18	72	10	Yes
M3	18	67	15	No
M3-X	18	67	15	Yes
M4	18	62	20	No
M4-X	18	62	20	Yes

**Table 2 membranes-13-00104-t002:** Hansen solubility parameters and boiling points of polybenzimidazole and solvents [37,38,39].

Materials	Hansen Solubility Parameter (MPa^0.5^)			R_HSP_ (Mpa^0.5^)	Boiling Point (°C)
	δ_d_	δ_p_	δ_h_	δ_S-P_	
PBI	20.4	6.6	7.5	-	-
DMAc	16.8	11.5	10.2	9.7	165
MeCN	15.3	18.0	6.1	16.0	82

**Table 3 membranes-13-00104-t003:** Comparison of OSN performance of pure ethanol permeance and solute rejection between PBI based membranes and commercial OSN membrane.

Membrane	Ethanol Permeance (L m^−2^ h^−1^ bar^−1^)	Solute	Rejection (%)	Ref.
PBI/MeCN-DBX	2.7	PPG (366 g/mol)	96.56	This work
MPF-50 ^a^	4.2	Raffinose (504 g/mol)	41	[48]
STAPMEM^TM^ 122 ^a^	2.4	Sudan black (456 g/mol)	94.1	[49]
DuraMem^®^ 300 ^a^	0.3	Methyl orange (327 g/mol)	94.5	[50]
PEI2K-GA	1.4	Methyl orange (327 g/mol)	89.6	[50]
PBI-PXDC	0.3	Crystal violet (408 g/mol)	75.7	[24]
PBI-50sPPSU-DBX-HPEI25k	4.24	Tetracycline (444 g/mol)	83	[51]
PBI-NGDGE	19	Methyl orange (327 g/mol)	14.12	[52]

^a^ Commercial OSN membranes.

## Data Availability

The data presented in this study are available on request from the corresponding author.

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
