# Peer review of "Improving the Separation Properties of Polybenzimidazole Membranes by Adding Acetonitrile for Organic Solvent Nanofiltration"

_membranes, 2023, doi:10.3390/membranes13010104_

Round 1

Reviewer 1 Report

Comments to the Author

This manuscript reported the strategy related to improving the separation properties of PBI membranes via using acetonitrile for OSN application. Moreover, the authors studied the rejection performance and stable permeances for various organic solvents. However, some basic studies are not included which the authors are advised to perform. The manuscript can be published in membranes in revised form after addressing some major corrections.

The authors should address the following concerns before submission.

1.     Morphology of the material surface plays a vital role in achieving high permeance and selectivity in rejection. Presence of acetonitrile has resulted in a dense and thick active layer. However, authors did not determine the roughness of the resulted membrane which can be easily measured from AFM studies.

2.     Authors are advice to include nanoindentation studies often use to determine hardness of the membrane.

3.     Author performed the separation studies and PPG rejections measured at 15 bar. Is there any particular reason of performing studies at 15 bar? Authors should provide the studies related to change in the permeance of solvent at different pressure.

4.     Long-term rejection and permeance studies are missing which are usually done to determine the long-term separation performance of the membranes.

5.     Has author observed any compaction of the membrane when performed OSN at 15 bar pressure?

6.     Has author noticed swelling of the membrane during chemical stability test?

Author Response

This manuscript reported the strategy related to improving the separation properties of PBI membranes via using acetonitrile for OSN application. Moreover, the authors studied the rejection performance and stable permeances for various organic solvents. However, some basic studies are not included which the authors must perform. The manuscript can be published in membranes in revised form after addressing some major corrections.

The authors should address the following concerns before submission.

We appreciate the reviewer's time and valuable comments. Please see below and the attached for additional figures and table.

1. Morphology of the material surface plays a vital role in achieving high permeance and selectivity in rejection. Presence of acetonitrile has resulted in a dense and thick active layer. However, authors did not determine the roughness of the resulted membrane which can be easily measured from AFM studies.

R1 : Thank you for the comment. The reviewer mentioned the surface morphology and roughness are important for OSN separation performance. However, the solvent permeance and rejection rely more on the pore size and the thickness of the active layer, which is the densest part on the top of the membrane, rather than the surface morphology. Therefore, this manuscript focused on cross-sectional structure changes (i.e., active layer thickness) depending on co-solvent. In addition, the active layer pore size of integrally skinned OSN membrane prepared by the phase separation is in the range from 0.5 – 2 nm resulting in a very smooth and flat surface.

2. Authors are advice to include nanoindentation studies often use to determine hardness of the membrane.

R2 : Thank you for the suggestions. For pressure driven filtration membranes, the hydraulic pressure tolerance is more important than the surface hardness to show the membrane stability at high pressure. We added the long-term stability test result in Figure 7 (c) and (d) with additional discussion. The readers can find the mechanical stability of the OSN membrane increased with MeCN contents resulting from the increase in the density and thickness of the active layer.

3. Author performed the separation studies and PPG rejections measured at 15 bar. Is there any particular reason of performing studies at 15 bar? Authors should provide the studies related to change in the permeance of solvent at different pressure.

R3 : We appreciate the reviewer’s comment. We performed the OSN test at different pressure (i.e., 5 bar, 10 bar and 15 bar). The pure ethanol flux increases with the applied pressure. The OSN test at 15 bar is an experiment under the harshest pressure condition that can be driven in our crossflow system. To observe the stability of the membrane in the harsh environment, the OSN tests in the study were conducted at 15 bar.

4. Long-term rejection and permeance studies are missing which are usually done to determine the long-term separation performance of the membranes.

R4 : We appreciate the reviewer’s good comment. The long-term stability test results were added as Figure 7(c) (M1-X) and (d) (M4-X) with additional discussion. The M4-X showed superior pressure tolerance compared to M1-X due to the dense membrane structure of M4-X.

5. Has author observed any compaction of the membrane when performed OSN at 15 bar pressure?

R5 : Thank you for the question. As shown in the membrane cross-section SEM image above, there is no noticeable decrease in membrane thickness after the OSN test at 15 bar after long term performance experiments (i.e., 96 hrs). However, a more severe permeance decline on M1-X than the M4-X is observed through the long-term stability test (added as Figure 7 (c), (d) in the manuscript). Based on this result, it can be assumed that M1-X was more affected by compaction under our operation condition (i.e., 15 bar) due to their less rigid structure, even though there are different parameters that effect the permeance decline such as fouling. Therefore it can be concluded that M4-X showed superior pressure tolerance by the dense membrane structure compared with M1-X.  

6. Has author noticed swelling of the membrane during chemical stability test?

R6 : Thank you for the question. We did the immersion test in acetone, methanol, isopropanol, ethanol, and toluene. The membranes were not swelled in organic solvents we tested, while the membrane swelled a bit in acetone and toluene.

Reviewer 2 Report

 The manuscript “Improving the separation properties of polybenzimidazole membranes by adding acetonitrile for organic solvent nanofiltration” is well written and technically sound. It can be considered for publication in Membrane’s journal after rectifying the the below comments.

  1. The PBI film had an evaporation time of 20 s before dipping in coagulation bath, on what bases was this time set. Also, was there any study carried out with different time interval like 30s and so on. This will give the authors exactly at what point the membrane achieves its optimum morphology.
  2. Secondly, how were the external condition maintained as the external temperature on the given day will vary the extent of evaporation (slower or faster) of the co-solvent.
  3. The novelty of the research should be highlighted in the introduction as there are numerous similar studies.
  4. The authors claim the fabricated membranes to have superior separation properties, superior to what? Give a comparison table with already published work in the recent years.

Author Response

Reviewer #2:

The manuscript “Improving the separation properties of polybenzimidazole membranes by adding acetonitrile for organic solvent nanofiltration” is well written and technically sound. It can be considered for publication in Membrane’s journal after rectifying the the below comments.

We appreciate the reviewer's time and valuable comments. Please see the responses below,

1. The PBI film had an evaporation time of 20 s before dipping in coagulation bath, on what bases was this time set. Also, was there any study carried out with different time interval like 30s and so on. This will give the authors exactly at what point the membrane achieves its optimum morphology.

R1 : Thank you for the comments and questions. As the evaporation time increases, the dope solution with low viscosity permeates more into the support, making it difficult to fabricate a uniform membrane. By giving an evaporation time of 20 seconds, solvent evaporation at the surface of the membrane occurred sufficiently and a uniform membrane was fabricated, and the explanation was added in the manuscript (lines 188-190).

2. Secondly, how were the external condition maintained as the external temperature on the given day will vary the extent of evaporation (slower or faster) of the co-solvent.

R2 : Thank you for the comments. We fabricated the membrane in an enclosed room with specific temperature and humidity (22 ℃ and 30-35 %) using an air-conditioner, and the explanation was added in the membrane fabrication section (lines 118-119).

3. The novelty of the research should be highlighted in the introduction as there are numerous similar studies.

R3 : We appreciate the reviewer’s comments. As the reviewer pointed out, there are many papers about PBI membranes fabricated with volatile co-solvents such as THF which has good affinity with PBI. But such co-solvent have low viscosity, resulting in reduced dope viscosity, which makes it difficult to prepare the mechanically robust membrane. To fabricate a PBI membrane with both better separation performance and mechanical stability, acetonitrile having a poor affinity with PBI was added into a low-concentration PBI dope solution. As a result, the viscosity of the PBI dope solution was increased and the dense OSN membranes were prepared by using acetonitrile as a co-solvent. An explanation was added in the introduction part.

4. The authors claim the fabricated membranes to have superior separation properties, superior to what? Give a comparison table with already published work in the recent years.

R4 : Thank you for the comment. We have added the comparison table (Table 3) and with additional discussion. In this manuscript, the separation performance was conducted using ethanol/PPG solution; thus, the OSN results using ethanol/solute in other papers were compared. Compared with other membranes, crosslinked PBI/MeCN in this study shows superior exclusion properties of low molecular weight solutes.

Reviewer 3 Report

Improving the separation properties of polybenzimidazole membranes by adding acetonitrile for organic solvent nanofiltration»

The authors of the article have developed highly efficient OSN membranes based on polybenzimidazole. To improve the properties of the membranes, the addition of acetonitrile (MeCN) to the dope solution was used, followed by crosslinking with dibromo-p-xylene. This article has a number of key advantages:

1.      The authors not only state changes in the morphology of membranes and their nanofiltration properties in case of using different dope solutions, but also explain in detail the observed changes in terms of the interaction of substances with each other, the viscosity of solutions, their volatility, etc.

2.      The membranes were tested not only with the use of model solutions, as is usually the case, but also a study was carried out on a real facility of the fine chemical industry (ecamsule in a toluene/methanol solution).  The efficiency of this recovery reached 90%.

Thus, this article may be of interest to a number of researchers involved in the issues of polymers, their properties, the development of membranes, as well as their introduction into industry.

The article is recommended for publication after minor revisions and clarifications.

1.      In Table 1, the designations "Y" and "N" probably correspond to the presence and absence of cross-linking with a DBX solution. Under the table or in the text, a transcript of the designations used should be placed.

2.      The text of the article talks a lot about crosslinking. It would be good if the authors demonstrated the mechanism of this crosslinking or at least gave a link to its description in the literature.

3.      Page 4, lines 149, 150. Equation (1) was used to calculate the permeability of membranes for both types of filtration, both dead-end and crossflow? Please add a clarification.

4.      4. Line 190. It is necessary to remove the formatting error that occurred when referring to table 2.

5.      Figure 4 shows that increasing the concentration of MeCN leads to an increase in rejection. At the moment, the M4 membrane has the best properties, with a MeCN concentration of 20 wt. %. Could a further increase in the concentration of MeCN lead to even higher rejection? In what concentration range is this regularity true? Or, perhaps, a further increase in the concentration of MeCN is irrational from the point of view of the solubility of substances or viscosity?

6.      In Figure 4C, it would be good to place the data on solvents in descending order of increasing their permeance. This will make it easier to visually perceive the presented data.

7.      In lines 311-313 it is said that the recovery of the ecamsul from toluene/methanol solution was difficult to carry out under crossflow filtrations due to the deposit of the powder. In this regard, dead-end filtration was used. Didn't this create even higher  problems? Wouldn't the deposited powder under dead-end filtration be more likely to contaminate the system and impede solvent permeation?

Author Response

Reviewer #3:

The authors of the article have developed highly efficient OSN membranes based on polybenzimidazole. To improve the properties of the membranes, the addition of acetonitrile (MeCN) to the dope solution was used, followed by crosslinking with dibromo-p-xylene. This article has a number of key advantages:

  1. The authors not only state changes in the morphology of membranes and their nanofiltration properties in case of using different dope solutions, but also explain in detail the observed changes in terms of the interaction of substances with each other, the viscosity of solutions, their volatility, etc.
  2. The membranes were tested not only with the use of model solutions, as is usually the case, but also a study was carried out on a real facility of the fine chemical industry (ecamsule in a toluene/methanol solution).  The efficiency of this recovery reached 90%.

Thus, this article may be of interest to a number of researchers involved in the issues of polymers, their properties, the development of membranes, as well as their introduction into industry.

The article is recommended for publication after minor revisions and clarifications.

We appreciate the reviewer's time and valuable comments. Please see below and the attached for additional figure.

1. In Table 1, the designations "Y" and "N" probably correspond to the presence and absence of cross-linking with a DBX solution. Under the table or in the text, a transcript of the designations used should be placed.

R1 : Thank you for the comment. We have modified Table 1 as the reviewer suggested.

2. The text of the article talks a lot about crosslinking. It would be good if the authors demonstrated the mechanism of this crosslinking or at least gave a link to its description in the literature.

R2 : We appreciate the reviewer’s suggestion. The illustration of the chemical crosslinking was newly added in Figure 5 (a) to show the mechanism and the chemical structure of the cross-linked PBI.

3. Page 4, lines 149, 150. Equation (1) was used to calculate the permeability of membranes for both types of filtration, both dead-end and crossflow? Please add a clarification.

R3 : We modified the sentence that equations (1) and (2) were used for both test methods.

4. Line 190. It is necessary to remove the formatting error that occurred when referring to table 2.

R4 : Thank you for the suggestion. We corrected the error.

5. Figure 4 shows that increasing the concentration of MeCN leads to an increase in rejection. At the moment, the M4 membrane has the best properties, with a MeCN concentration of 20 wt. %. Could a further increase in the concentration of MeCN lead to even higher rejection? In what concentration range is this regularity true? Or, perhaps, a further increase in the concentration of MeCN is irrational from the point of view of the solubility of substances or viscosity?

R5 : By decreasing PBI solubility in the mixed solvent with more MeCN addition, PBI didn’t dissolve in a solvent including 25 wt% MeCN. Therefore, the PBI dope solution was prepared with a solvent containing up to 20 wt% of MeCN, and the explanation was added in the manuscript (lines 185-187)

6. In Figure 4C, it would be good to place the data on solvents in descending order of increasing their permeance. This will make it easier to visually perceive the presented data.

R6 : Thank you for the comment. We think Figure 4 (c) reviewer mentioned in the question is Figure 7 (c). We have modified the figure of organic solvent permeance (Figure 7 (e)) as the reviewer suggested.

7. In lines 311-313 it is said that the recovery of the ecamsul from toluene/methanol solution was difficult to carry out under crossflow filtrations due to the deposit of the powder. In this regard, dead-end filtration was used. Didn't this create even higher problems? Wouldn't the deposited powder under dead-end filtration be more likely to contaminate the system and impede solvent permeation?

R7 : We agree with reviewer’s comment. The deposit of the ecamsule on the membrane in the dead-end system is more severe rather than the cross-flow system. To select the best membranes we fabricated in this study, the ecamsule solution test was performed using a dead-end system. In a dead-end system, the ecamsule solution was stirred using a magnetic bar in the dead-end system to minimize the deposit of ecamsule on a membrane. After confirming the ecamsule separation performance of the cross-linked PBI/MeCN membrane, the membrane with good separation properties was selected and experimented with using a cross-flow system. The ecamsule nanofiltration test was conducted using a cross-flow system with an enlarged cross-linked PBI/MeCN membrane (5.0 m2) on a pilot scale, and the ecmasule solution permeance increased 2.7 times. As mentioned, it is confirmed that the deposited ecamsule under a dead-end system impedes solvent permeation.

Round 2

Reviewer 1 Report

Comments to the Author

This manuscript reported the strategy related to improving the separation properties of PBI membranes via using acetonitrile for OSN application. Authors have improved the manuscript by including some fundamental experiments such as long term permeance and variation in the solvent permeance with increasing pressure, etc.

The manuscript can be published in membranes in the present form.

Reviewer 2 Report

The authors have answered all the comments with sufficient clarity and this paper can be accepted for publication in “Membranes” journal.

However, the authors are advised to write the explanation in easy-to-understand language and check the manuscript for errors throughout.

e.g. In this study, we demonstrated that a PBI membrane with both better separation performance and mechanical stability may be fabricated through co-solvent acetonitrile addition and crosslinking.

Corrected: In this study, we demonstrated that a PBI membrane with both better separation performance and mechanical stability can be fabricated through co-solvent (acetonitrile) addition and crosslinking.